# Data-Efficient Model Learning for Control with Jacobian-Regularized Dynamic-Mode Decomposition

**Brian E. Jackson**
Robotics Institute
Carnegie Mellon University
`brianjackson@cmu.edu`

Jeong Hun Lee
Robotics Institute
Carnegie Mellon University
`jeonghunlee@cmu.edu`

Kevin Tracy
Robotics Institute
Carnegie Mellon University
`ktracy@cmu.edu`

Zachary Manchester
Robotics Institute
Carnegie Mellon University
`zacm@cmu.edu`

**Abstract:** We present a data-efficient algorithm for learning models for model-predictive control (MPC). Our approach, Jacobian-Regularized Dynamic-Mode Decomposition (JDMD), offers improved sample efficiency over traditional Koopman approaches based on Dynamic-Mode Decomposition (DMD) by leveraging Jacobian information from an approximate prior model of the system, and improved tracking performance over traditional model-based MPC. We demonstrate JDMD's ability to quickly learn bilinear Koopman dynamics representations across several realistic examples in simulation, including a perching maneuver for a fixed-wing aircraft with an empirically derived high-fidelity physics model. In all cases, we show that the models learned by JDMD provide superior tracking and generalization performance within a model-predictive control framework, even in the presence of significant model mismatch, when compared to approximate prior models and models learned by standard Extended DMD (EDMD).

## 1 Introduction

In recent years, both model-based optimal-control [1, 2, 3, 4] and data-driven reinforcement-learning methods [5, 6, 7] have demonstrated impressive success on complex, nonlinear robotic systems. However, both approaches suffer from inherent drawbacks: Data-driven methods often require extremely large amounts of data and fail to generalize outside of the domain or task on which they were trained. On the other hand, model-based methods require an accurate model of the system to achieve good performance. In many cases, high-fidelity models can be too difficult to construct from first principles or too computationally expensive to be of practical use. However, low-order approximate models that can be evaluated cheaply at the expense of controller performance are often available. With this in mind, we seek a middle ground between model-based and data-driven approaches in this work.

We propose a method for learning bilinear Koopman models of nonlinear dynamical systems for use in model-predictive control that leverages derivative information from an approximate prior dynamics model of the system in the training process. Given the increased availability of differentiable simulators [8, 9], this approximate derivative information is readily available for many systems of interest. Our new algorithm builds on Extended Dynamic Mode Decomposition (EDMD), which learns Koopman models from trajectory data [10, 11, 12, 13, 14], by adding a derivative regularization term based on derivatives computed from a prior model. We show that this new algorithm, Jacobian-regularized Dynamic Mode Decomposition (JDMD), can learn models with dramatically fewer samples than EDMD, even when the prior model differs significantly from the true dynamics

6th Conference on Robot Learning (CoRL 2022), Auckland, New Zealand.

of the system. We also demonstrate the effectiveness of these learned models in a model-predictive control (MPC) framework. The result is a fast, robust, and sample-efficient pipeline for quickly training a model that can outperform MPC controllers using both approximate analytical models as well as models learned using traditional Koopman approaches.

Our work is most closely related to the recent works of Folkestad et. al. [13, 15, 16], which learn bilinear models and apply nonlinear model-predictive control directly on the learned bilinear dynamics. Other recent works have combined linear Koopman models with model-predictive control [12] and Lyapunov control techniques with bilinear Koopman models [17]. Our contributions are:

- A novel extension to Dynamic Mode Decomposition, called JDMD, that incorporates gradient information from an approximate analytic model
- A recursive, batch QR algorithm for solving the least-squares problems that arise when learning bilinear dynamical systems using DMD-based algorithms, including JDMD and EDMD

The remainder of the paper is organized as follows: In Section 2 we provide some background on the application of Koopman operator theory to controlled dynamical systems and review some related works. Section 3 then describes the proposed JDMD algorithm. In Section 4 we outline a memory-efficient technique for solving the large, sparse linear least-squares problems that arise when applying JDMD and other DMD-based algorithms. Section 5 then provides simulation results and analysis of the proposed algorithm applied to control tasks on a cartpole, a quadrotor, and a small foam airplane with an experimentally determined aerodynamics model, all subject to significant model mismatch. We also discuss the suitability of the conventional open-loop prediction error as a metric for evaluating dynamics model used in closed-loop control frameworks. In Section 6 we discuss the limitations of our approach, followed by some concluding remarks in Section 7.

## 2 Background and Related Work

### 2.1 Koopman Operator Theory

The theoretical underpinnings of the Koopman operator and its application to dynamical systems has been extensively studied [11, 18, 19, 20, 21, 22]. Rather than describe the theory in detail, we highlight the key concepts employed by the current work and refer the reader to the existing literature on Koopman theory for further details.

We start by assuming a controlled, nonlinear, discrete-time dynamical system,

$$x^+ = f(x, u), \tag{1}$$

where $x \in \mathcal{X} \subseteq \mathbb{R}^{N_x}$ is the state vector, $u \in \mathbb{R}^{N_u}$ is the control vector, and $x^+$ is the state at the next time step. Assuming the dynamics are control affine, the nonlinear finite-dimensional system (1) can be represented *exactly* by an infinite-dimensional bilinear system through the Koopman canonical transform [22]. This bilinear Koopman model takes the form,

$$y^+ = Ay + Bu + \sum_{i=1}^{m} u^i C^i y = g(y, u), \tag{2}$$

where $y = \phi(x)$ is a nonlinear mapping from the finite-dimensional state space $\mathcal{X}$ to the infinite-dimensional Hilbert space of *observables* $\mathcal{Y}$. In practice, we approximate (2) by restricting $\mathcal{Y}$ to be a finite-dimensional vector space, in which case $\phi$ becomes a finite-dimensional nonlinear function of the state variables that can be either chosen heuristically based on domain expertise or by learning [23, 24, 25].

Intuitively, $\phi$ "lifts" our state $x$ into a higher dimensional space $\mathcal{Y}$ where the dynamics are approximately bilinear, effectively trading dimensionality for bilinearity. Similarly, we can perform an "unlifting" operation by projecting a lifted state $y$ back into the original state space $\mathcal{X}$. In this work,

since we embed the original state within the nonlinear mapping [11, 15, 26, 27, 28], $\phi$ is constructed in such a way that this unlifting is linear:

$$x = Gy. \tag{3}$$

We note that our proposed method does not require this assumption: any mapping could be used. The problem of finding an optimal mapping is itself a major area of research, and many recent studies have focused on jointly learning both the model and the mapping [23, 24, 25, 29, 30]. While clearly advantageous, learning an optimal mapping is not the focus of this paper. Instead, we focus on incorporating prior information from an approximate model in a way that is applicable to any lifting function, and we rely on simple mappings that are chosen heuristically in all of our examples.

## 2.2 Extended Dynamic Mode Decomposition

A lifted bilinear system of the form (2) can be learned from $P$ samples of the system dynamics $(x_j^+, x_j, u_j)$ using Extended Dynamic Mode Decomposition (EDMD) [15, 21]. We first define the following data matrices:

$$Z_{1:P} = \begin{bmatrix} y_1 & y_2 & \cdots & y_P \\ u_1 & u_2 & \cdots & u_P \\ u_1^1 y_1 & u_2^1 y_2 & \cdots & u_P^1 y_P \\ \vdots & \vdots & \ddots & \vdots \\ u_1^m y_1 & u_2^m y_2 & \cdots & u_P^m y_P \end{bmatrix}, \quad Y_{1:P}^+ = \begin{bmatrix} y_1^+ & y_2^+ & \cdots & y_P^+ \end{bmatrix}, \tag{4}$$

where $u_k^i$ is the $i$-th element of the control vector at time $k$. We then concatenate all of the model coefficient matrices from (2) as follows:

$$E = \begin{bmatrix} A & B & C^1 & \cdots & C^m \end{bmatrix} \in \mathbb{R}^{N_y \times N_z}. \tag{5}$$

The model learning problem can then be written as the following linear least-squares problem:

$$\underset{E}{\text{minimize}} \left\| E Z_{1:P} - Y_{1:P}^+ \right\|_2^2 \tag{6}$$

EDMD is closely related to classical feature-based machine learning approaches like the "kernel trick" used in support vector machines [31], but extends these ideas to bilinear models of controlled dynamical systems.

# 3 Jacobian-Regularized Dynamic Mode Decomposition

We now present JDMD as a straightforward adaptation of the original EDMD algorithm described in Section 2.2. Given $P$ samples of the dynamics $(x_j^+, x_j, u_j)$, and an approximate discrete-time dynamics model,

$$x^+ = \tilde{f}(x, u), \tag{7}$$

we can evaluate the Jacobians of our approximate model $\tilde{f}$ at each of the sample points: $\tilde{A}_j = \frac{\partial \tilde{f}}{\partial x}, \tilde{B}_j = \frac{\partial \tilde{f}}{\partial u}$. After choosing a nonlinear mapping $\phi : \mathbb{R}^{N_x} \mapsto \mathbb{R}^{N_y}$ our goal is to find a bilinear dynamics model (2) that matches the Jacobians of our approximate model, while also matching our dynamics samples. We accomplish this by penalizing differences between the Jacobians of our learned bilinear model with respect to the original states $x$ and controls $u$, and the Jacobians we expect from our analytical model. These *projected Jacobians* are calculated by differentiating through the *projected dynamics*:

$$x^+ = G \left( A\phi(x) + Bu + \sum_{i=1}^m u^i C^i \phi(x) \right) = \bar{f}(x, u). \tag{8}$$

Differentiating (8) with respect to $x$ and $u$ gives us

$$\bar{A}_j = \frac{\partial \hat{f}}{\partial x}(x_j, u_j) = G \left( A + \sum_{i=1}^m u_j^i C^i \right) \Phi(x_j) = GE\hat{A}(x_j, u_j) = GE\hat{A}_j, \tag{9a}$$

$$\bar{B}_j = \frac{\partial \hat{f}}{\partial u}(x_j, u_j) = G \Big( B + \begin{bmatrix} C^1 x_j & \cdots & C^m x_j \end{bmatrix} \Big) = GE\hat{B}(x_j, u_j) = GE\hat{B}_j, \tag{9b}$$

where $\Phi(x) = \partial\phi/\partial x$ is the Jacobian of the nonlinear map $\phi$, and

$$\hat{A}(x,u) = \begin{bmatrix} I_{N_y} \\ 0 \\ u^1 I_{N_y} \\ u^2 I_{N_y} \\ \vdots \\ u^m I_{N_y} \end{bmatrix} \Phi(x) \in \mathbb{R}^{N_z \times N_x}, \quad \hat{B}(x,u) = \begin{bmatrix} 0 \\ I_{N_u} \\ [\phi(x)\ 0\ ...\ 0] \\ [0\ \phi(x)\ ...\ 0] \\ \vdots \\ [0\ 0\ ...\ \phi(x)] \end{bmatrix} \in \mathbb{R}^{N_z \times N_u}. \quad (10)$$

We then solve the following linear least-squares problem:

$$\underset{E}{\text{minimize}} \ \ (1-\alpha)\big\|EZ_{1:P} - Y_{1:P}^+\big\|_2^2 + \alpha \sum_{j=1}^{P} \left( \big\|GE\hat{A}_j - \tilde{A}_j\big\|_2^2 + \big\|GE\hat{B}_j - \tilde{B}_j\big\|_2^2 \right). \quad (11)$$

Problem (11) has $(N_y + N_x^2 + N_x \cdot N_u) \cdot P$ rows and $N_y \cdot N_z$ columns. Given that the number of rows in this problem grows quadratically with the state dimension, solving it can be computationally challenging. The next section proposes an algorithm to address this challenge without resorting to a distributed-memory system. This solution method also has the benefit of allowing incremental updates to the bilinear system, enabling online model learning.

## 4  Efficient Recursive Least Squares

In its canonical formulation, a linear least squares problem can be represented as the following unconstrained optimization problem:

$$\min_x \|Fx - d\|_2^2. \quad (12)$$

We assume $F$ is a large, sparse matrix and that solving it directly using a QR or Cholesky decomposition requires too much memory for a single computer. While solving (12) using an iterative method such as LSMR [32] or LSQR [33] is possible, we find that these methods do not work well in practice for solving (11) due to ill-conditioning. Standard recursive methods for solving these problems are able to process the rows of the matrices sequentially to build a QR decomposition of the full matrix, but also tend to suffer from ill-conditioning [34, 35, 36].

To overcome these issues, we propose an alternative recursive method based. We solve (12) by dividing rows of $F$ into batches:

$$F^T F = F_1^T F_1 + F_2^T F_2 + \ldots + F_N^T F_N. \quad (13)$$

The main idea is to maintain and update an upper-triangular Cholesky factor $U_i$ of the first $i$ terms of the sum (13). Given $U_i$, we can calculate $U_{i+1}$ using the QR decomposition, as shown in [37]:

$$U_{i+1} = \sqrt{U_i^T U_i + F_{i+1}^T F_{i+1}} = \text{QR}_\text{R}\left( \begin{bmatrix} U_i \\ F_{i+1} \end{bmatrix} \right), \quad (14)$$

where $\text{QR}_\text{R}$ returns the upper triangular matrix $R$ from the QR decomposition. For an efficient implementation, this function should be an "economy" or "Q-less" QR decomposition since the $Q$ matrix is never needed.

We also handle regularization of the normal equations, equivalent to adding quadratic or Tikhonov regularization to the original least squares problem, during the base case of our recursion,

$$U_1 = \text{QR}_\text{R}\left( \begin{bmatrix} F_1 \\ \sqrt{\lambda}I \end{bmatrix} \right), \quad (15)$$

where $\lambda$ is a scalar regularization weight. To ensure fair comparisons, the results presented in the next section for both EDMD and JDMD correspond to the best-performing $\lambda$ values found by sweeping over a wide parameter range.

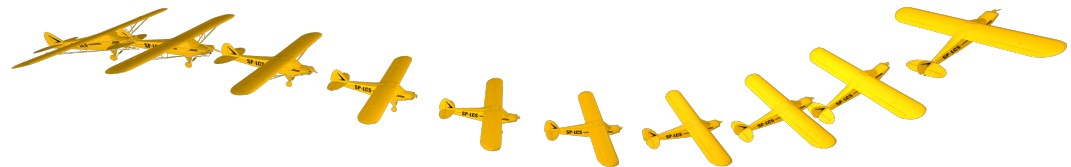

(a) Expert demonstration of a high angle-of-attack perching maneuver that minimizes velocity at the goal position with complex, post-stall aerodynamic forces.

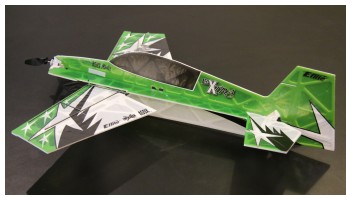

(b) E-Flite AS3Xtra airplane model used in hardware data collection.

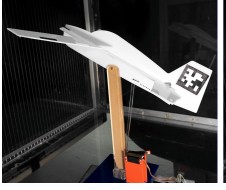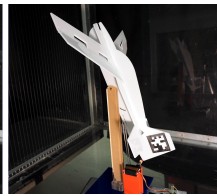

(c) Wind-tunnel experimental setup for collecting aerodynamic data [38].

Figure 1: Complex dynamics of a perching fixed-wing airplane. High-angle-of-attack perching maneuvers (top) require the modeling of complex post-stall aerodynamic effects. The simulated aerodynamic forces were modeled using flight data collected from real-world hardware experiments (bottom) [38].

## 5   Experimental Results

This section presents the results of several simulation experiments to evaluate the performance of JDMD. We specify two models for each simulated system: a *nominal* model, which is simplified and contains both parametric and non-parametric model error, and a *true* model, which is used exclusively for simulating the system and evaluating algorithm performance.

All models were trained by simulating the "true" system with a nominal controller to collect data in the region of the state space relevant to the task. A set of fixed-length trajectories were collected, each at a sample rate of 20-25 Hz. The bilinear EDMD model was trained using the same approach introduced by Folkestad and Burdick [15]. When applying MPC to the learned Koopman models, the projected Jacobians (9) were used, since this projected system is much more likely to be controllable than the lifted one and reduces the computational complexity of the MPC controller. This results in a nonlinear model in the original state space, which is linearized about the reference trajectory to create a linear MPC controller. All continuous-time dynamics were discretized with an explicit fourth-order Runge-Kutta method. Code for all experiments is available at https://github.com/bjack205/BilinearControl.jl.

### 5.1   Systems and Tasks

**Cartpole:** We perform a swing-up task on a cartpole system. The *true* model includes Coulomb friction between the cart and the floor, viscous damping at both joints, and a deadband in the control input that were not included in the *nominal* model. Additionally, the mass of the cart and pole model were altered by 20% and 25% with respect to the nominal model, respectively. The following nonlinear mapping was used when learning the bilinear models: $\phi(x) = [\,1,\,x,\,\sin(x),\,\cos(x),\,\sin(2x),\,\sin(4x),\,T_2(x),\,T_3(x),\,T_4(x)\,] \in \mathbb{R}^{33}$, where $T_i(x)$ is a Chebyshev polynomial of the first kind of order $i$. All reference trajectories for the swing up task were generated using ALTRO [37, 39].

**Quadrotor:** We track point-to-point linear reference trajectories from various initial conditions on a full 3D quadrotor model. The *true* model includes aerodynamic drag terms not included in the *nominal* model, as well as parametric error of roughly 5% on the system parameters (e.g. mass, rotor arm length, etc.). The model was trained using a nonlinear mapping of $\phi(x) =$

$[1,\ x,\ T_2(x),\ \sin(p),\ \cos(p),\ R^T v,\ v^T R R^T v,\ p \times v,\ p \times \omega,\ \omega \times \omega] \in \mathbb{R}^{44}$, where $p$ is the quadrotor's position, $v$ and $\omega$ are the translational and angular velocities respectively, and $R$ is the rotation matrix representation of the quadrotor's attitude.

**Airplane:** We perform a post-stall perching maneuver on a high-fidelity model of a fixed-wing airplane. The perching trajectory is produced using trajectory optimization (see Figure 1a). Perching involves flight at high angles of attack, where the aerodynamic lift and drag forces are extremely complex and difficult to model from first principles. We leverage previous works that fit post-stall aerodynamics models using empirical data from wind-tunnel experiments [38, 40]. The *true* model includes these empirical nonlinear flight dynamics [38], while the *nominal* model uses a simple flat-plate wing model with linear lift and quadratic drag coefficient approximations. The bilinear models use a nonlinear mapping $\phi \in \mathbb{R}^{68}$, which includes the aircraft attitude (expressed as a vector of Modified Rodriguez Parameters [41]), powers of the angle of attack and side slip angle, the body frame velocity, various cross products with the angular velocity, and 3rd and 4th order Chebyshev polynomials of the states.

## 5.2 Sensitivity to Model Mismatch

While a significant amount of model mismatch is introduced in all examples, a natural argument against model-based methods is that they are only as good as the model's ability to capture the salient dynamics of the system. Therefore, we investigated the effect of increasing model mismatch by incrementally increasing the Coulomb friction coefficient $\mu$ between the cart and the floor for the cartpole stabilization task (recall the nominal model assumed zero friction). The results are shown in Table 1. As expected, the number of training trajectories required to find a good stabilizing controller increases with $\mu$. We achieved the results above by setting $\alpha = 0.01$, corresponding to a decreased confidence in our model, thereby placing greater weight on the experimental data. The standard EDMD approach always required more samples, and was unable to find a good enough model above friction values of 0.4. While this could likely be remedied by adjusting the nonlinear mapping $\phi$, the proposed approach works well with the given basis. Note that the nominal MPC controller failed to stabilize the system above friction values of 0.1, so again, we demonstrate that we can improve MPC performance substantially with just a few training samples by combining analytical derivative information and data sampled from the true dynamics.

| $\mu$ | Nominal | EDMD | JDMD |
|---|---|---|---|
| 0.0 | ✓ | 3 | 2 |
| 0.1 | ✓ | 19 | 2 |
| 0.2 | ✗ | 6 | 2 |
| 0.3 | ✗ | 15 | 2 |
| 0.4 | ✗ | ✗ | 3 |
| 0.5 | ✗ | ✗ | 7 |
| 0.6 | ✗ | ✗ | 12 |

Table 1: Training trajectories required to stabilize the cartpole with given friction coefficient.

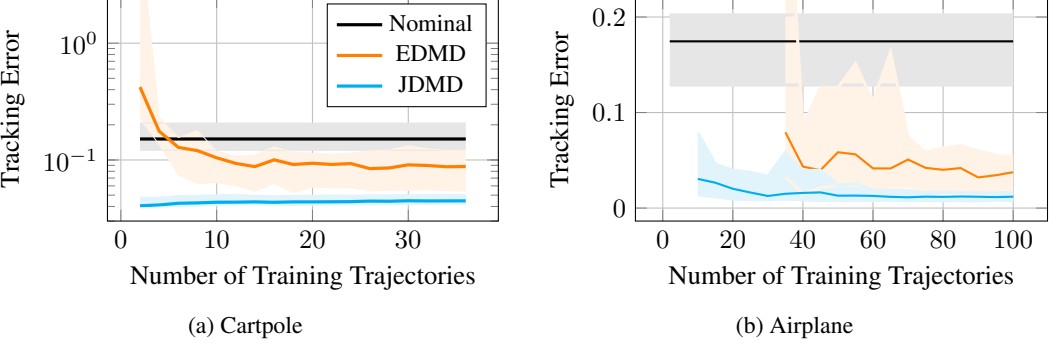

(a) Cartpole  (b) Airplane

Figure 2: MPC tracking error vs training trajectories for both the cartpole (left) and airplane (right). Tracking error is defined as the average L2 error over all the test trajectories between the reference and simulated trajectories. The median error is shown as a thick line, while the shaded regions represent the 5% to 95% bounds on the 10 test trajectories.

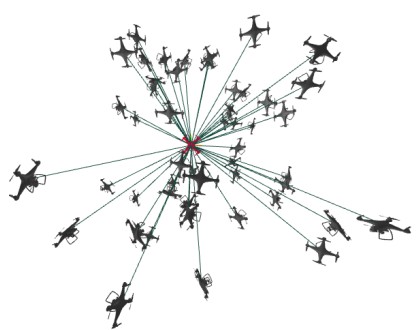 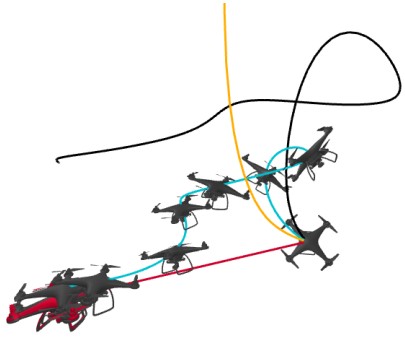

(a) Point-to-point trajectories and initial conditions for testing MPC on a 6-DOF quadrotor.

(b) Closed-loop trajectories of nominal MPC (black), EDMD (orange), and JDMD (cyan) for tracking a dynamically infeasible, point-to-point trajectory (red).

Figure 3: Visualizations of the tests on the full quadrotor model.

## 5.3 Sample Efficiency

We compare the sample efficiency of several algorithms on the cartpole swing-up task in Figure 2. As shown, JDMD achieves the best performance overall, and does so with only two training trajectories. In comparison, traditional EDMD requires about 10 iterations to achieve consistent performance. Similar results were obtained for the airplane perching example (see Figure 2b), where EDMD requires over three times the number of samples (35 vs 10) compared to JDMD and never achieves the same closed-loop performance.

## 5.4 Generalization

We demonstrate the generalizability of JDMD on a quadrotor. The task is to return to the origin, given an initial condition sampled from a uniform distribution centered at the origin. Test initial conditions are sampled from a distribution larger than that of the training data. Given the goal of tracking a straight line back to the origin, we test 50 initial conditions, many of which are far from the goal, have large velocities, or are nearly inverted (see Figure 3a). The results using an MPC controller are shown in

|  | **Nominal** | **EDMD** | **JDMD** |
|---|---|---|---|
| Success Rate | **82%** | 18% | 80% |
| Median | 0.30 | 0.63 | **0.11** |
| 5% Quantile | 0.13 | 0.08 | **0.03** |
| 95% Quantile | 0.38 | 2.62 | **0.23** |

Table 2: Performance summary of MPC tracking of 6-DOF quadrotor. Other than success rate, all values are the tracking error of the successfully stabilized trajectories.

Table 2, demonstrating the generalizability of JDMD, given that the algorithm was only trained on 30 initial conditions sampled relatively sparsely given the size of the sampling window. EDMD only successfully brings about 18% of the samples to the origin, while the majority of the time resulting in trajectories like those in Figure 3b. JDMD improves the tracking performance of nominal MPC, which is subject to a constant-bias error due to model mismatch, as shown in Figure 3b.

## 5.5 Model Prediction Error vs. Controller Performance

Much of the previous literature focuses on open-loop prediction error for evaluating learned-dynamics models [11, 12, 15, 19, 22, 42]. While intuitive, we argue that this is a poor metric when the end goal is closed-loop control performance. As shown in the histogram of open-loop prediction error in Figure 4a, the open-loop prediction error of JDMD (trained with 8 trajectories) is significantly higher over 100 test trajectories, with 74% of tests resulting in a prediction error of approximately 1.5 compared to 25% for EDMD (trained with 24 trajectories). Despite worse open-loop prediction performance, the JDMD model outperforms the EDMD model in closed-loop

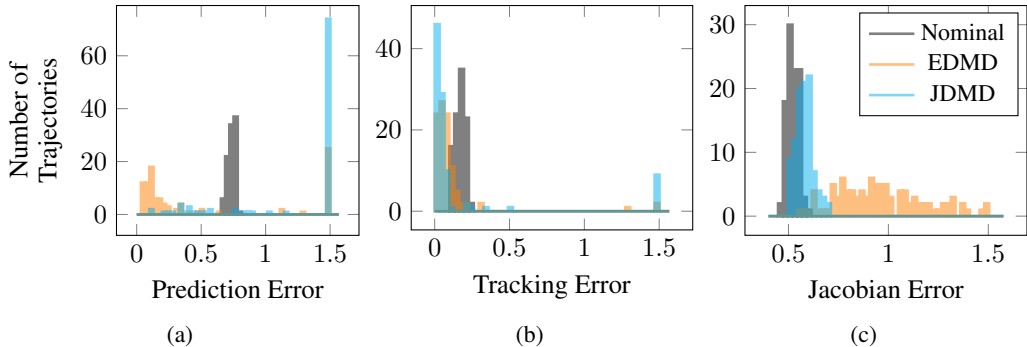

Figure 4: Histograms displaying prediction, tracking, and Jacobian error over 100 test trajectories with randomly sampled initial conditions for the airplane perching problem. The EDMD model is trained with 24 trajectories and the JDMD model is trained with 8. While JDMD has poor open-loop prediction (a), it has better closed-loop tracking performance (b) and better matching of the dynamics Jacobians (c).

tracking (see Figure 4b). Given that MPC, like most closed-loop controllers, relies on the behavior of the model under small perturbations (i.e. derivative information), the difference in tracking performance may be explained by JDMD achieving a distribution with much lower Jacobian error than EDMD (see Figure 4c). This suggests that open-loop prediction error is not necessarily a good metric for evaluating models that will be used in control applications, and that models sufficient for closed-loop control may be learned with far less data.

## 6 Limitations

Many of the limitations of the proposed approach derive from the limitations of Koopman approaches more broadly, such as the sensitivity to the nonlinear mapping selected and the limitation to control-affine continuous dynamics. While the presented single rigid-body systems such as the quadrotor or airplane have similar dimensionality to many autonomous systems of interest, extensions to systems with many degrees of freedom may be difficult computationally, given that Jacobian matrices grow in size with the square of the state dimension. As with most data-driven techniques, it is difficult to claim that our method will increase performance in all cases. It is possible that having an extremely poor prior model may hurt rather than help the training process, especially if the derivative information from the approximate model has the incorrect sign.

## 7 Conclusions and Future Work

We have presented JDMD, a simple but powerful extension to EDMD that incorporates derivative information from an approximate prior model. We have tested JDMD in combination with a simple linear MPC control policy across a range of systems and tasks, and have found that the resulting combination can dramatically increase sample efficiency over EDMD, often improving over a nominal MPC policy with just a few sample trajectories. We also argued that the conventional open-loop dynamics prediction error is a poor metric for evaluating models used in closed-loop control frameworks. Substantial areas for future work remain: most notably, demonstrating the proposed pipeline on hardware. Additional directions include applications on sytems with many degrees of freedom such as those whose dynamics are governed by discretized PDEs, online learning or adaptive control applications, combining simulated and real data through the use of modern differentiable physics engines [9, 8], residual dynamics learning, as well as the development of specialized numerical methods for solving nonlinear optimal control problems using the learned bilinear dynamics.

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
