# OpenReview forum: "Data-Efficient Model Learning for Control with Jacobian-Regularized Dynamic-Mode Decomposition"
_robot-learning.org/CoRL/2022/Conference — CoRL 2022 Poster_

### Official Review · Reviewer_5W5c · 2022-07-12

**Originality:** Very Good
**Technical Quality:** Very Good
**Clarity Of Presentation:** Very Good
**Impact:** 4

**Recommendation:**

Weak Accept: I recommend accepting the paper, but will not argue for my recommendation if the majority of other reviewers have a different opinion.

**Summary:**

The paper proposed a method for regularised fitting of Koopman models based on use of prior model knowledge introduced to the algorithm through Jacobians. In doing so, its main contribution is to extend the Extended Dynamic Mode Decomposition (EDMD) approach for model predictive control (MPC) applications. The performance of the approach is evaluated in a series of experiments for the control of three simulated benchmark problems, and shows improved performance as compared to traditional MPC or EDMD.

**Issues:**

- Please discuss the plausibility of the assumption made in equation 3 and its implications for practical use.
- Please repeat all evaluations for multiple trials with randomised data sets and provide statistics on the results. As a minumum I would expect the mean+/-standard devation performance to be presented.
- Please fix the formatting problem on Lines 182-184.

**Quality Of The Limitations Section:**

Limitations are not well addressed

**Reviewer Expertise:**

4: The reviewer is confident but not absolutely certain that the evaluation is correct

**Robotics Focus:**

Highly relevant to robotics but no hardware experiments

**Strengths And Weaknesses:**

Strengths:
- The paper is very clearly written. The quality of the figures is excellent.
- The mathematical development is appropriate.
- The evaluations provide a thoughtful analysis of the proposed approach (beyond simply claiming that it works better than the state of the art) and go a good way toward characterising how it performs and why it behaves the way it does.
- The apparent performance gains are impressive.
- The video is excellent in explaining the work.

Weaknesses:
- Evaluations are only presented for a single trial, so it is impossible to judge whether the results are representative or statistically significant.
- The assumption that the inverse mapping is linear (equation 3) seems very strong, and not applicable in practice.
- No results are presented for a physical system, causing doubt as to the practical applicability.
- The discussion of limitations is brief and more or less denies that there are any. This is implausible.

**Summary Of Recommendation:**

Overall, the paper is of good quality, and addresses an important problem. It presents an elegant and seemingly effective approach to the problem in hand that I would expect to perform well if applied to a real, physical system. I did not detect any issues with the mathematical development. The evaluations are thoughtful and engaging. The results appear to be impressive, but lack statistical rigour making their reliability dubious.

---

> ### Author Response · Authors · 2022-08-22
> **Author Response**
>
> Dear Reviewer,
>
> Thank you for the generous and detailed feedback; we firmly believe that it has made our paper stronger as a result. We have done our best to address the concerns raised, which we detail below. We have attached a revised draft of the paper, with changes marked in blue.
>
> ## Issue 1: Evaluations presented for a single trial
> First, we want to clarify that our results are over many trials, and that the reported results were the mean error over these trials. We have added percentile bounds on all of the applicable plots, adjusted the plots to report the median instead of the mean, and now specify the number of test samples for each of the analyses. This was a significant oversight in the original draft and wholeheartedly agree with the reviewer that this was a weakness of the original draft. We believe this addition adds important information for each of the experiments. We also want to clarify that our sampling was, in general, very generous in that we sampled over wide uniform distributions where the final trajectories varied significantly, rather than small perturbations about a single trial. Please see the attached code for the specifics on the exact methodology we used in each of the studies.
>
> ## Issue 2: Linear inverse mapping assumption
> We have added clarification in the paper that our usage of a linear unlifting map G is not a critical assumption of our methodology, and was done only for simplicity and convenience. This is accomplished simply by including the original state vector in the lifted state (i.e. by including an identity mapping in $\phi$), enabling a trivial and exact linear inverse mapping. We adopted this approach based on previous work in the literature and found it to work well in practice, but any mapping could be used with our approach.
>
> ## Issue 3: Limitations section is too limited/brief
> Based on reviewer feedback, we have expanded the limitations section to include additional limitations associated with learning Koopman models, in addition to specifics of our study. This includes the sensitivity of Koopman models to the selection of the nonlinear mapping $\phi$, Jacobian regularization’s adoption in larger, more computationally-intensive systems, and the study’s translation to hardware. We hope that the limitations section now provides more insight and scope regarding the limitations of our algorithm and study.
>
> ## Issue 4: No hardware demonstration
> While we agree that the paper would benefit from hardware experiments, we are unable to perform such experiments at the present time. However, we believe that the airplane perching task, which uses a high-fidelity full-flight-envelope aerodynamics model built from wind-tunnel data, is a good representation of our algorithm’s performance on a challenging and highly nonlinear real-world control problem. We have added additional information to the paper about this model’s derivation and accuracy. We also apply a high degree of model mismatch between the nominal and “true” simulated models in all experiments, which gives confidence that our technique will translate well to future hardware deployments.

---

> > ### Comment · Reviewer_5W5c · 2022-08-25
> > **Reveiwer response to author replies and revisions**
> >
> > This reviewer would like to thank the authors for their detailed efforts in responding to my comments. The improvements to the paper substantially improve my evaluation of the work, and as a result I will be raising my score to strong accept.

---

### Official Review · Reviewer_4PJQ · 2022-07-13

**Originality:** Good
**Technical Quality:** Good
**Clarity Of Presentation:** Very Good
**Impact:** 2

**Recommendation:**

Weak Reject: I recommend rejecting the paper, but will not argue for my recommendation if the majority of other reviewers have a different opinion.

**Summary:**

The proposed approach improves the sample efficiency of Koopman-based models by introducting an auxillary loss minimizing the difference between the linearized Koopman dynamics and a linearized approximate model, coined Jacobian regularization. The method is evaluated using a downstream trajectory tracking task using MPC, on cartpole, quadcopter and plane dynamics in simulation.

**Issues:**

I refer to my question above on the need for Koopman models given the feature regression it is essentially doing if the true state is in the latent state.

**Quality Of The Limitations Section:**

Limitations are addressed clearly

**Reviewer Expertise:**

3: The reviewer is fairly confident that the evaluation is correct

**Robotics Focus:**

Highly relevant to robotics but no hardware experiments

**Strengths And Weaknesses:**

I think ideas of combining black-box models with physics-based inductive biases is an interesting research direction. Moreover, using about derivative or Jacobian information is a seldom-explored direction in machine learning. I thought this paper was also clearly written and had a very pleasing quality to the presentation.

My main issue with the paper is its motivation and execution. This is a 'solution-first' paper rather than a 'problem-first' one, as Koopman models are part of the motivation. I took issue with this beacuse Koopman dynamical systems are rather niche, and I have not seen convincing evidence that they are a particularly promising direction of research. Nonlinearities with infinite taylor series require infinitely-sized latent spaces, and therefore Koopman dynamical systems are best suited to systems with polynomial nonlinearities like Van der Pol oscillators, whereas robotics typically trignometric nonlinearities.  Moreover, compared to direct function approximation of the dynamical system, Koopman dynamical systems introduce three approximators (encoder, dynamics and decoder) which means there are additional sources of error and drift. I believe this paper is missing a baseline of a standard 2 layer MLP (perhaps with sine and cosine features) that approximates the dynamics directly. For the trajectory tracking MPC task, this MLP can also be linearized offline using forward-mode automatic differentiation.

My second issue is with the hand-crafted latent space. Since the dynamic state $x$ is part of the latent state $\phi(x)$, this suggests that the Koopman model is already predicting the next state directly using feature regression, i.e. $x_{t+1} = \mathbf{w}^T\phi(x_t)$? The fact of the Koopman model having a linear decoder was very strange to me, and suggests this model is infact something simpler, since previous Koopman methods i have seen use some form of neural network autoencoder. I think this feature regression baseline should also be used a baseline, since it is also straightforward to learn and linearize.

Next, while the Jacobian regularization is an interesting one, the paper does not really engage with the general question of 'how should I use an approximate oracle model?' and rather proposes one solution. I think the paper could be improved if it evaluates alternative approaches as well, such as learning a residual model, or augmenting the dataset with noise and encouraging predictive similarity when predicting outside of the data distribution. Without this investigation, the paper is improving on vanilla Koopman models by providing some form of oracle knowledge, which is not so insightful as the advantage is clear.

Regarding the evaluation, I found the trajectory tracking task on simple environments like cartpole a bit lacking for a venue like CoRL, especially as there are no real-world experiments. Moreover, there seems to be a complete absence of reporting confidence intervals and evaluating over random seeds.

**Summary Of Recommendation:**

The submission is highly focused around one particular solution, and does not engage outside Koopman models or other methods of using approximate oracle models. I have concerns over the motivation for Koopman, and I find the experiments a bit limited in complexity and statistical evaluation. Therefore, I am leaning towards rejection.

---

> ### Author Response · Authors · 2022-08-22
> **Author Response**
>
> Dear Reviewer,
>
> Thank you for the generous and detailed feedback; we firmly believe that it has made our paper stronger as a result. We have done our best to address the concerns raised, which we detail below. We have attached a revised draft of the paper, with changes marked in blue, in our response to the meta review.
>
> ## Issue 1: Koopman as motivation
> We agree with the author that Koopman theory does come with limitations, which we have noted more clearly in the limitations section. The limitations include the crafting of the nonlinear mapping into the lifted space as well as the inverse mapping. In this work we do not cover learning of the embedding functions of the nonlinear mapping. We agree with the reviewer that a framework in which this mapping is learned along with the model is very advantageous, and some recent work has focused on this [1,2,3,4]. However, we feel that addressing this is orthogonal to the main thrust of paper, which we now note in the background section of the paper. To support this, we added an example using an MLP to learn the dynamics instead of Koopman approaches, and found similar improvements by leveraging the proposed approach of leveraging derivative information from an approximate model within the learning process. Based on this, we believe the proposed approach can be readily applied within previously proposed methods that learn the embedding function alongside the model, and that similar advantages will be observed.
>
>
> ## Issue 2: Inverse linear mapping
> We have added clarification in the paper that our usage of a linear unlifting map G is not a critical assumption of our methodology, and was done only for simplicity and convenience. This is accomplished simply by including the original state vector in the lifted state (i.e. by including an identity mapping in $\phi$), enabling a trivial and exact linear inverse mapping. We adopted this approach based on previous work in the literature and found it to work well in practice, but any mapping could be used with our approach.
>
> ## Issue 3: Lack of a baseline comparison to an MLP
> We have added an example substituting an MLP for a Koopman model to the paper, and showed that the proposed approach of incorporating Jacobian information into the loss function is advantageous there, as well. We believe this adds an insightful baseline comparison to the paper.
>
> ## Issue 4: No hardware demonstration
> While we agree that the paper would benefit from hardware experiments, we are unable to perform such experiments at the present time. However, we believe that the airplane perching task, which uses a high-fidelity full-flight-envelope aerodynamics model built from wind-tunnel data, is a good representation of our algorithm’s performance on a challenging and highly nonlinear real-world control problem. We have added additional information to the paper about this model’s derivation and accuracy. We also apply a high degree of model mismatch between the nominal and “true” simulated models in all experiments, which gives confidence that our technique will translate well to future hardware deployments.
>
> ## Issue 5: Environment examples are simple
> We have added clarification on the complexity of the dynamics in our chosen systems. We agree that the cartpole and quadrotor systems are common, low-dimensional systems. These systems are meant to act as benchmark systems that still allow for intuition and thorough analysis and that becomes prohibitive with larger systems. For this reason, we have also included the high-fidelity perching airplane example, which is a significantly more complex and challenging system to control. While perching may appear trivial, it involves flight at high angle-of-attack, where the post-stall aerodynamic effects are extremely sensitive, complex, and hard to model from first principles. As stated in response to Issue 4, the simulator dynamics involve high-fidelity fits to real wind-tunnel experiments, and we have cited previous works that provide more details on the complexity of this problem. All of our examples are also subject to significant model mismatch, which suggests that extensions to hardware are reasonable.

---

> > ### Author Response · Authors · 2022-08-22
> > **Author Response (cont.)**
> >
> > ## Issue 6: Absence of statistical results
> > We have added percentile bounds on all of the applicable plots, as well as specifying the number of test samples for each of the analyses. This was a significant oversight in the original draft, which only reported the mean error over all the samples, and we believe this addition adds more important information for each of the experiments. We also want to clarify that our sampling was, in general, very generous in that we sampled over wide uniform distributions where the final trajectories varied significantly, rather than small perturbations about a single trial. Please see the attached code for the specifics on the exact methodology we used in each of the studies.
> >
> > References:
> >
> > [1] Folkestad, Carl, Skylar X. Wei, and Joel W. Burdick. "KoopNet: Joint Learning of Koopman Bilinear Models and Function Dictionaries with Application to Quadrotor Trajectory Tracking." 2022 International Conference on Robotics and Automation (ICRA). IEEE, 2022.
> >
> > [2] Kaiser, Eurika, J. Nathan Kutz, and Steven L. Brunton. "Data-driven discovery of Koopman eigenfunctions for control." Machine Learning: Science and Technology 2.3 (2021): 035023.
> >
> > [3] Li, Qianxiao, et al. "Extended dynamic mode decomposition with dictionary learning: A data-driven adaptive spectral decomposition of the Koopman operator." Chaos: An Interdisciplinary Journal of Nonlinear Science 27.10 (2017): 103111.
> >
> > [4] Wang, Rongyao, Yiqiang Han, and Umesh Vaidya. "Deep koopman data-driven optimal control framework for autonomous racing." Early Access 5 (2021).

---

> > ### Comment · Reviewer_4PJQ · 2022-08-24
> > **Feature regression issue**
> >
> > This rebuttal statement does not address my feature regression comment.
> >
> > The latent state $\phi$ includes the actual state $x$ (c.f. line 165), that means that $\phi(x_{t+1}) = A \phi(x_t)$ contains a linear model $x_{t+1} = W \phi(x_t)$. If this feature regression model is sufficiently accurate, then the Koopman machinery isn't needed to model the dynamical system. Can the authors comment on the performance of this model? This is why the linear inverse mapping is so strange, if the latent state can include the true state then I don't understand why Koopman theory is needed.

---

> > > ### Author Response · Authors · 2022-08-25
> > > **Feature Regression Clarification**
> > >
> > > We agree with the reviewer regarding the linearity of the prediction model for an autonomous (i.e. not controlled) dynamical system in the latent Koopman feature space, and the close relationship between Koopman models and classical feature-based regression in machine learning. We’ve added additional language at the end of the EDMD background to clarify this point. This linearity is, in fact, the main attraction of Koopman models. We would like to clarify one critical point: While the state prediction model for an autonomous system can be written linearly in $\phi$, as the reviewer suggests, the systems we are dealing with are controlled, and therefore the Koopman model is nonlinear (bilinear) in the latent state and control vectors. Our work leverages this bilinear structure throughout. We would also like to point out that including the original state vector as a feature in $\phi$ is standard in the literature on Koopman models [1, 2, 3, 4, 5], and that it does not limit the applicability of our method to models that omit this feature. We have added clarification of this point to the paper in the background and related works section.
> > >
> > > We've attached an updated paper to the meta review.
> > >
> > > ## References:
> > > [1] D. Bruder, X. Fu, and R. Vasudevan, “Advantages of bilinear Koopman realizations for the modeling and control of systems with unknown dynamics,” IEEE Robotics and Automation Letters, vol. 6, no. 3, pp. 4369–4376, 2021.
> > >
> > > [2] C. Folkestad and J. W. Burdick, “Koopman NMPC: Koopman-based learning and nonlinear model predictive control of control-affine systems,” in 2021 IEEE International Conference on Robotics and Automation (ICRA), 2021, pp. 7350–7356.
> > >
> > > [3] B. Huang, X. Ma, and U. Vaidya, “Feedback stabilization using koopman operator,” in 2018 IEEE Conference on Decision and Control (CDC), 2018, pp. 6434–6439.
> > >
> > > [4] X. Ma, B. Huang, and U. Vaidya, “Optimal quadratic regulation of nonlinear system using koopman operator,” in 2019 American Control Conference (ACC), 2019, pp. 4911–4916.
> > >
> > > [5] G. Mamakoukas, M. Castano, X. Tan, and T. Murphey, “Local Koopman operators for data-driven control of robotic systems,” 2019.

---

> > ### Comment · Reviewer_4PJQ · 2022-08-24
> > **MLP Baseline**
> >
> > Thanks for adding this experiment. How is this MLP trained? It does not look correct to me that performance oscillates as the dataset is increased. I would expect that performance should improve with dataset size (like your EDMD result in figure 2a). Are you training the MLP till convergence? Are you using a validation set to prevent overfitting?

---

> > > ### Author Response · Authors · 2022-08-25
> > > **MLP Clarification**
> > >
> > > Great questions. With respect to the training for the MLP, we used standard mini-batch stochastic gradient descent with ADAM and a fixed learning rate. We used cross-validation testing (separate from our test set) and stopped the optimization if the validation loss started increasing consistently. The reason the closed-loop performance doesn’t improve with increasing samples in Fig 2b is that closed-loop performance is not the metric we’re training for. The standard instantaneous model prediction error we use in training doesn’t guarantee the model will work well in an MPC controller, which relies on the local gradient information. To demonstrate this we've added Figure 4, which instead plots the loss function (both training and test), and demonstrates the reviewer’s expected behavior of improved performance with an increasing number of training trajectories. We believe this is a critical insight: while our learned model is getting better and better at predicting the instantaneous discrete dynamics, this doesn’t correlate to improved closed-loop control performance, and point we also make in Section 5.5. We appreciate the feedback and feel this has made the paper stronger.
> > >
> > > We've attached an updated paper to the meta review.

---

### Official Review · Reviewer_Uwdc · 2022-07-30

**Originality:** Good
**Technical Quality:** Very Good
**Clarity Of Presentation:** Very Good
**Impact:** 4

**Recommendation:**

Weak Accept: I recommend accepting the paper, but will not argue for my recommendation if the majority of other reviewers have a different opinion.

**Summary:**

Motivated by Koopman operator theory, this paper considers the setting where a given nonlinear dynamical system is approximated by a lifted bilinear system, where the parameters of the latter are fit on (state, control, next-state) triplets with respect to both next-state prediction loss as well as a dynamics Jacobian matching loss. The paper sketches efficient least squares implementations for this approach. Experimental results show that adding a Jacobian loss term improves MPC tracking error; that learning is more data efficient; leads to better generalization to initial conditions as well as greater robustness to model mismatch.



**Issues:**

- A plot of performance as a function of alpha would be insightful to add to understand the relative important of Jacobian matching and also sensitivity of performance.

- How are fixed basis functions phi and unlifting matrix G to be chosen for a new problem?



**Quality Of The Limitations Section:**

Limitations are addressed clearly

**Reviewer Expertise:**

5: The reviewer is absolutely certain that the evaluation is correct and very familiar with the relevant literature

**Robotics Focus:**

Highly relevant to robotics but no hardware experiments

**Strengths And Weaknesses:**

Strengths:

- This paper reminded the reviewer of a recent approach called TaSIL: Taylor Series Imitation Learning (https://arxiv.org/abs/2205.14812) which experimentally and theoretically shows the benefit of Jacobian matching in addition to vanilla action-prediction matching in Imitation Learning settings. One can view the proposed approach in this paper from a similar lens where the original nonlinear dynamics is an expert that the lifted bilinear dynamics attempts to imitate. It would be interesting to see if TaSIL analysis applies in this setting or not.

- The paper is clearly written and the experiments are nicely organized and convincing.

Weaknesses:

- No real on-robot demonstrations.
- Eqn 3 seems like an overly strong assumption
- Learning is partial in the sense that it only covers the coefficient matrices of Eqn 2, not the embedding function phi or the unlifting matrix G. It is not clear how to set these effectively.
- While effective in practice, the Projected bilinear MPC is a heuristic without a sound theoretical justification.
- The empirical results are on smallish systems - it is not clear how far such an approach can be pushed.


**Summary Of Recommendation:**

Nice, clear formulation that leads to an effective dynamics learning and MPC approach, with several empirical gains albeit demonstrated in smallish systems (and not on real hardware).

---

> ### Author Response · Authors · 2022-08-22
> **Author Response**
>
> Dear Reviewer,
>
> Thank you for the generous and detailed feedback; we firmly believe that it has made our paper stronger as a result. We have done our best to address the concerns raised, which we detail below. We have attached a revised draft of the paper with changes marked in blue in our response to the meta review.
>
> ## Issue 1: Plot of performance as a function of alpha
> We have added an additional plot showcasing the performance of both open-loop model rollouts and MPC tracking error over a range of alpha values. Unsurprisingly, high alpha values do result in high model prediction error (open-loop performance). Interestingly, MPC maintains good tracking performance over the same range of alpha values despite high prediction error; this raises an interesting question regarding the effectiveness of prediction error as a metric to evaluate models for closed-loop controls tasks. We see this as an interesting topic that requires separate, future research that is beyond the scope of this paper.
>
> ## Issue 2: How are basis functions phi and unlifting matrix G chosen?
> In this work we do not cover learning of the embedding function phi. We agree with the reviewer that a framework in which this mapping is learned along with the model is very advantageous, and much recent work has focused on this problem [1,2,3,4]. However, we feel that addressing this issue is beyond the scope of our paper. We have added further acknowledgements in the paper that selecting a good embedding function is a known issue with Koopman approaches, and that learning them is an area of active research. We chose to avoid addressing this directly in the current work, and leave it open to future work, since we feel the advantages of the proposed approach should apply, regardless of the particular embedding. In our specific examples, the basis functions were chosen to include common trigonometric functions and orthogonal (e.g. Chebyshev) polynomials.
>
> In regards to the unlifting matrix G, we embed the original state vector within the lifted state (i.e. we include an identity mapping in $\phi$). Therefore, the unlifting becomes linear, and we have noted this in the revised paper. We have added clarification in the paper that our usage of a linear unlifing map is not a critical assumption of our methodology, and is only done for simplicity and convenience. We found it to work well in practice, but any mapping could be used in principle.
>
> ## Issue 3: Projected bilinear MPC is a heuristic
> Several reviewers suggested that the discussion on projected versus lifted MPC was confusing. We agree that this detracted from the central focus of the paper, so we have decided to remove these sections from the paper, opting instead for a small explanation and justification at the beginning of the results section. We hope this provides better focus to the paper and reduces confusion. We do believe there is a solid theoretical backing for the projected MPC approach, as it simply reduces to standard nonlinear MPC in the original state space. Projecting back into the original state space also makes it easy to enforce constraints on the original states and controls, and significantly reduces the computational cost.

---

> > ### Author Response · Authors · 2022-08-22
> > **Author Response (cont.)**
> >
> > ## Issue 4: Empirical results only on smallish systems
> > We have added clarification on the complexity of the dynamics in our chosen systems, despite their small scale. We agree that the cartpole and planar quadrotor systems are common, small systems that act as baselines that have been used as the sole model of interest in previously published papers on this subject [5,6]. For this reason, we have included the perching airplane model to involve more complex dynamics in our study. While perching may appear trivial, it involves flight at high angle-of-attacks, where the post-stall aerodynamic effects are extremely sensitive, complex, and hard to model from first principles. The dynamics involve high-order polynomial approximations fitted to wind-tunnel experiments, and we have cited previous works that provide more details on the complexity of this problem. We have also noted in the limitations section that scaling up to systems with many degrees of freedom might be a limitation, given that derivative information scales quadratically with the state dimension. However, we believe the proposed recursive QR approach should mitigate this in practice.
> >
> > ## Issue 5: No hardware demonstration
> > While we agree that the paper would benefit from hardware experiments, we are unable to perform such experiments at the present time. However, we believe that the airplane perching task, which uses a high-fidelity full-flight-envelope aerodynamics model built from wind-tunnel data, is a good representation of our algorithm’s performance on a challenging and highly nonlinear real-world control problem. We have added additional information to the paper about this model’s derivation and accuracy. We also apply a high degree of model mismatch between the nominal and “true” simulated models in all experiments, which gives confidence that our technique will translate well to future hardware deployments.
> >
> > ## References:
> >
> > [1] Folkestad, Carl, Skylar X. Wei, and Joel W. Burdick. "KoopNet: Joint Learning of Koopman Bilinear Models and Function Dictionaries with Application to Quadrotor Trajectory Tracking." 2022 International Conference on Robotics and Automation (ICRA). IEEE, 2022.
> >
> > [2] Kaiser, Eurika, J. Nathan Kutz, and Steven L. Brunton. "Data-driven discovery of Koopman eigenfunctions for control." Machine Learning: Science and Technology 2.3 (2021): 035023.
> >
> > [3] Li, Qianxiao, et al. "Extended dynamic mode decomposition with dictionary learning: A data-driven adaptive spectral decomposition of the Koopman operator." Chaos: An Interdisciplinary Journal of Nonlinear Science 27.10 (2017): 103111.
> >
> > [4] Wang, Rongyao, Yiqiang Han, and Umesh Vaidya. "Deep koopman data-driven optimal control framework for autonomous racing." Early Access 5 (2021).
> >
> > [5] Folkestad, Carl, and Joel W. Burdick. "Koopman NMPC: Koopman-based learning and nonlinear model predictive control of control-affine systems." 2021 IEEE International Conference on Robotics and Automation (ICRA). IEEE, 2021.
> >
> > [6] Folkestad, Carl, et al. "Extended dynamic mode decomposition with learned koopman eigenfunctions for prediction and control." 2020 american control conference (acc). IEEE, 2020.

---

### Official Review · Reviewer_R4eq · 2022-07-31

**Originality:** Good
**Technical Quality:** Fair
**Clarity Of Presentation:** Good
**Impact:** 2

**Recommendation:**

Weak Reject: I recommend rejecting the paper, but will not argue for my recommendation if the majority of other reviewers have a different opinion.

**Summary:**

Combining prior knowledge with small data model learning is a significant problem for robust Model Predictive Control. The authors proposed a novel model learning method, JDMD that adds the Jacobian information of the dynamical system to Dynamic Mode Decomposition, DMD. Because DMD treats a nonlinear system as a linear system, the nonlinear control problem can be solved as the linear control. The proposed method devised the efficient computation algorithm for model learning. In addition, the authors demonstrated the validity of sample efficiency of the gradient information-based DMD by trajectory tracking problem of three numerical simulations, Cartpole, Airplane, and, Quadrotor.

**Issues:**

1. Model Error Evaluation\
See above
2. detail prior Jacobian knowledge \
How did the prior Jacobian knowledge defined in each experiment?
3. I need more description to clarify the difference between Lifted and Projected MPC.\
The MPC of the previous EDMD[13] is solved in the observables space $\mathcal{Y}$. Is the state space where the QP problem solves a difference?
4. Does the $\phi(x)$ term appear in $\hat{B}$ of equation (10)?\
I think differentiation of equation (8) with respect to $u_i$ gives us the term of $C_i\phi(x)$.


**Quality Of The Limitations Section:**

Limitations are not well addressed

**Reviewer Expertise:**

4: The reviewer is confident but not absolutely certain that the evaluation is correct

**Robotics Focus:**

Highly relevant to robotics but no hardware experiments

**Strengths And Weaknesses:**

Strength
1. Jacobian-Regularized DMD (JDMD)\
JDMD may be so robust than the full data-driven learning that this method can reflect the dynamical property a designer knows. The efficient recursive algorithm based on the QR decomposition is also proposed.
1. Experimental results show that JDMD improves the Quadratic Problem to track the reference trajectory. \
This paper evaluated sample efficiency for model learning in Sec. 6.2, extrapolation data performance in Sec. 6.3, and bias error of prior knowledge in Sec. 6.5 comparing with the non-gradient information model learning. Every result indicates the proposed method improves the MPC score.

Weaknesses

1. There is no model error evaluation.\
The model error between true dynamics and the estimated model is interesting to confirm the proposed method is superior. Furthermore, The Jacobians matrix error among true, prior, and estimated would assess the performance of the proposed algorithm.
1. The limitation of the target dynamics system and ambiguous design of the prior knowledge\
The Conventional bilinear Koopman Model[13] assumes that the dynamical system is the Control-affine, but this paper defines the more general form as shown in Equation (1). This assumption is critical to lift the dynamical system onto the bilinear model[Surana, 2016]. A detailed description is necessary to clarify the limitation of the JDMD. On the other hand, obtaining the dynamical system Jacobians sometimes is a challenge, even if the dynamics model has already been received. The clear guide of the prior design is significant for applying the wide system.

Surana, Amit. "Koopman operator based observer synthesis for control-affine nonlinear systems." 2016 IEEE 55th Conference on Decision and Control (CDC). IEEE, 2016.


**Summary Of Recommendation:**

I consider the limitations of the proposed method to apply broad dynamical systems. JDMD method needs control affine system and the Jacobian knowledge, and the discussion seems not enough about these demands. Furthermore, because this paper presently concentrates on evaluating MPC score, the model error of prediction state and Jacobians should study to indicate the JDMD supremacy.

---

> ### Author Response · Authors · 2022-08-22
> **Author Response**
>
> Dear Reviewer,
>
> Thank you for the generous and detailed feedback; we firmly believe that it has made our paper stronger as a result. We have done our best to address the concerns raised, which we detail below. We have attached a revised draft of the paper with changes marked in blue in our response to the meta review.
>
>
> ## Issue 1: Model error evaluation
> We have added an additional study showcasing the model prediction error of the estimated bilinear dynamics, and how it translates to controller performance. We have demonstrated that good agreement can be achieved for open-loop rollouts over a range of regularization (alpha) values. Interestingly, when high model error is apparent, the closed-loop MPC performance of the Jacobian-regularized Koopman model does not appear to be greatly affected. This raises an interesting question regarding the effectiveness of prediction error as a metric to evaluate models for closed-loop control tasks. We see this as an interesting topic that requires separate, future research. For this paper specifically, we would like to highlight that the focus of the current work is primarily learning dynamics models for closed-loop control, and that open-loop dynamics prediction error or error in the learned Jacobians are imperfect metrics, whereas controller performance is the metric we care most about.
>
> ## Issue 2.1 Limitation of bilinear Koopman representation for broad application in dynamics:
> We agree with the author that the conventional bilinear Koopman model assumes the dynamics to be control-affine, thus acting as a limitation specific to the Koopman theory. We added language describing in the paper. However, we do believe that Jacobian regularization can be useful beyond Koopman frameworks (i.e. JDMD). Therefore, we have also added an example replacing the Koopman model with a neural network, where Jacobian regularization still improves control performance. We hope this demonstrates that Jacobian regularization can extend beyond the limitations of the bilinear Koopman model.
>
> ## Issue 2.2 Prior Jacobian knowledge
> Obtaining derivative information for many systems of interest has been facilitated by recent developments in differentiable simulators, as now noted in the paper. The Jacobian information required by the proposed method is made readily available by these tools. We also stress that the benefits of Jacobian regularization are maintained even in the presence of relatively large model mismatch (the subject of Section 5.4), such that only an approximate model is needed.
>
> ## Issue 3: Lifted vs Projected MPC
> Several reviewers suggested that the discussion on projected versus lifted MPC was confusing. We agree that this detracted from the central focus of the paper, so we decided to remove these sections from the paper, opting instead for a small explanation and justification at the beginning of the results section. We hope this provides better focus to the paper and reduces confusion. To clarify, the projected MPC QP problem in our work is solved in the original state space. Projecting back into the original state space makes it easy to enforce constraints on the original states and controls, and significantly reduces computational cost. Enforcing the implicit constraints imposed by the nonlinear mapping within the MPC controller is nontrivial, and neglecting them results in a linear system that is almost certainly uncontrollable, so we believe projecting back into the original state space is well-justified.
>
> ## Issue 4: Error in Eq (10)
> We have fixed the error in eq (10), which erroneously had x instead of $\phi(x)$ in the matrix for the $\hat{B}$ term. Thank you for catching this.
>
> ## Issue 5: No hardware demonstration
> While we agree that the paper would benefit from hardware experiments, we are unable to perform such experiments at the present time. However, we believe that the airplane perching task, which uses a high-fidelity full-flight-envelope aerodynamics model built from wind-tunnel data, is a good representation of our algorithm’s performance on a challenging and highly nonlinear real-world control problem. We have added additional information to the paper about this model’s derivation and accuracy. We also apply a high degree of model mismatch between the nominal and “true” simulated models in all experiments, which gives confidence that our technique will translate well to future hardware deployments.

---

> > ### Comment · Reviewer_R4eq · 2022-08-26
> > **Model Error Evaluation**
> >
> > Dear Authors,
> >
> > Thank you for submitting a revised paper with additional study and description.
> > The latest version has been substantially corrected and cleared up almost questions.
> > Therefore, my comment is focused on the generalization performance of model learning.
> >
> > ### Issue 1: Model error evaluation
> > I agree that a perfect model is not necessary but a just one is enough for MPC. However, I believe that evaluating the model and Jacobian error is significant in validating JDMD performance. The large model error may become the closed-loop system by MPC unstable. JDMD introduces the Jacobian-regularized term to prevent the dynamics model degradation brought by small data uncertainties. However, I am concerned with that only example of Sec. 5.5 cannot confirm the advance of the model learning by Jacobian-regularization. Although this is an excessive guess, L2-norm regularization may improve the generalization performance of model learning. Actually, Fig. 3b in the previous version showed that L2-norm regularized EDMD has a superior tracking error for fraction $\gamma < 2$ than JDMD. Therefore, to validate the JDMD, I infer that generalization model error should be independently evaluated before assessing the MPC control performance. Because authors in detail confirmed the MPC performance, my consideration is what generalizes the dynamic model.
> >
> > To evaluate generalization performance, it should be indicated that the JDMD model error is superior to EDMD in the test data. I agree that the open-loop dynamics prediction error is an unsuitable criterion for unstable system identification. Therefore, the prediction error of the closed-loop ought to be compared. Although the prediction error must evaluate under a “fixed” controller for strictness, it would be no problem if the performance of the controller and model has a strong relationship.
> >
> > Even if the EDMD and JDMD have an unstable open-loop system, the author's idea can be demonstrated by Jacobian evaluation. For instance, true, prior, and estimated Jacobian can is calculated, such as the cart pole in Sec. 5.4. Then, by comparing the Jacobian error between JDMD with EDMD, it can be validated that JDMD has superior generalization performance. If the Jacobian-regularized brings about the generalization of model learning, the estimated Jacobian error would be smaller than EDMD or nominal (prior).
> >
> > ### Minor questions are below.
> >
> > Are the labels in the Fig. 6a legend reversed?
> >
> > Dash line graph labeled as “Closed loop” in Fig. 6a seems to lack the points due to the system divergence. However, the caption said, “The missing open-loop values are points there the states of the open-loop system diverged to infinity.”

---

> > > ### Author Response · Authors · 2022-08-26
> > > **Author Response**
> > >
> > > Dear Reviewer,
> > >
> > > Thank you for your additional feedback. We will do our best to address them below:
> > >
> > > ## Issue 1: Model error evaluation
> > > First, we would like to clarify that there was a typo regarding Fig. 3b in the original submission: the EDMD labels are incorrect - the thin orange line actually corresponds to L2-regularized EDMD (λ = 0.1) and the thick line corresponds to unregularized EDMD (λ = 0.0). Fig. 3a correctly corresponds to the legend in the original submission. This was an oversight of the original submission, and we apologize for any confusion that this has caused. We would also like to clarify that the λ values presented in the plots (λ = 0.1 for EDMD, and λ = 10^-5 for JDMD) and throughout the paper were determined from a sweep over a range of λ values to ensure that the best L2-regularization values were used for each algorithm. The revised paper only presents the EDMD and JDMD results corresponding to these tuned-λ values to reflect what would be done in practice. We have now made additional note of this in the paper at the end of Section 4. The addition of L2-regularization does improve the generalizability of EDMD, but still results in worse performance than L2-regularized JDMD overall. This gives us confidence that Jacobian-regularization itself is dramatically improving both performance and generalizability, not just L2-regularization.
> > >
> > > Regarding the generalizability of the model, we have added the requested plot of Jacobian error for the nominal, EDMD, and JDMD models in Fig 4b in the newly revised paper. Unsurprisingly, JDMD learns the Jacobians provided by the nominal model, while EDMD has significantly more error in the Jacobians, which, as the reviewer suggested, is likely one of the reasons EDMD doesn’t generalize as well as JDMD, as demonstrated by the results in Sections 5.3 and 5.4. We have also added discussion in Section 5.5 regarding JDMD’s superior minimization of Jacobian error, and how that may translate to better closed-loop controls performance.
> > >
> > > ## Issue 2: Are the labels in the Fig. 6a legend reversed?
> > > We fixed this error in Fig 6a, which had the “open-loop” and “closed-loop” labels flipped.
> > >
> > > We've attached an updated paper to the meta review.

---

### Author Response · Authors · 2022-08-22
**Response to Meta Review**

Dear Reviewer,

Thank you for the compiled feedback; we firmly believe that the reviewers’ feedback has made our paper stronger as a result. We have done our best to address the concerns raised, which we detail below. We have attached a revised draft of the paper with changes marked in blue.

## Issue 1: No hardware evaluation is provided
While we agree that the paper would benefit from hardware experiments, we are unable to perform such experiments at the present time. However, we believe that the airplane perching task, which uses a high-fidelity full-flight-envelope aerodynamics model built from wind-tunnel data, is a good representation of our algorithm’s performance on a challenging and highly nonlinear real-world control problem. We have added additional information to the paper about this model’s derivation and accuracy. We also apply a high degree of model mismatch between the nominal and “true” simulated models in all experiments, which gives confidence that our technique will translate well to future hardware deployments.

## Issue 2: The evaluations are performed on very simple tasks with a single trial
We have added clarification on the complexity of the dynamics in our chosen systems, despite their small scale. We agree that the cartpole and quadrotor systems are common, small systems that act as baselines. For this reason, we have included the perching airplane model to involve more complex dynamics in our study. While perching may appear trivial, it involves flight at high angle-of-attacks, where the post-stall aerodynamic effects are extremely sensitive, complex, and hard to model from first principles. As stated in response to Issue 1, the dynamics involve high-order polynomial approximations fitted to wind-tunnel experiments, and we have cited previous works that provide more details on the complexity of this problem.

We have also added percentile bounds on all of the applicable plots, as well as specifying the number of test samples for each of the analyses. This was a significant oversight in the original draft, which only reported the mean error over all samples. We believe this addition adds more important information for each of the experiments. We also want to clarify that our sampling was, in general, very generous in that we sampled over wide uniform distributions where the final trajectories varied significantly, rather than small epsilon perturbations about a single trial. Please see the attached code for the specifics on the exact methodology we used in each of the studies.

## Issue 3: The assumption of equations 3 (linear unlifting) is overly strong
We have added clarification in the paper that our usage of a linear unlifting map G is not a critical assumption of our methodology, and was done only for simplicity and convenience. This is accomplished simply by including the original state vector in the lifted state (i.e. by including an identity mapping in $\phi$), enabling a trivial and exact linear inverse mapping. We adopted this approach based on previous work in the literature and found it to work well in practice, but any mapping could be used with our approach. We have added discussions regarding this in both the background and limitations section of the paper.

## Issue 4: Effect of modeling error should be discussed
We have added an additional study showcasing the model prediction error of the estimated bilinear dynamics, and how it translates to controller performance. We have demonstrated that good agreement can be achieved for open-loop rollouts over a range of regularization (alpha) values. Interestingly, when high model error is apparent, the closed-loop MPC performance of the Jacobian-regularized Koopman model does not appear to be greatly affected. This raises an interesting question regarding the effectiveness of prediction error as a metric to evaluate models for closed-loop control tasks. We see this as an interesting topic that requires separate, future research. For this paper specifically, we would like to highlight that the focus of the current work is primarily learning dynamics models for closed-loop control, and that open-loop dynamics prediction error or error in the learned Jacobians are imperfect metrics, whereas controller performance is the metric we care most about. In addition, we would like to clarify that in our original submission, we added a high degree of model mismatch (i.e. modeling error, model disagreement) between the nominal and “true” simulated models in all experiments, and that the effects of model mismatch were explicitly studied.

---

> ### Author Response · Authors · 2022-08-22
> **Revised Submission**
>
> Revised PDF, with changed marked in blue.

---

> ### Author Response · Authors · 2022-08-25
> **Revised Submission (2)**
>
> Attached is the latest revision, incorporating some additional feedback from Reviewer 4PJQ.

---

> ### Author Response · Authors · 2022-08-26
> **Revised Submission (3)**
>
> Attached is the latest revision, incorporating some additional feedback from Reviewer R4eq.

---

### Meta-Review · Area_Chair_43Cr · 2022-08-08

**Recommendation:** Accept (Poster)
**Confidence:** 4

**Metareview:**

### Initial Meta-review:
#### Strength:

This paper is clearly written and provides mathematically sound formulation. The ideas presented in this paper appear to be an interesting research direction. The results are nicely organized and convincing.

#### Weakness:

No hardware evaluation is provided. The evaluations are performed on very simple tasks with a single trial. Statistical evaluation is highly suggested with multiple trials. The assumption of equations 3 appears to be overly strong, whose practical plausibility should be discussed. In addition, effect of modelling error should be discussed.

### Final Meta-review:
The authors have addressed the questions and concerns raised by the reviewers and revised the paper in a detailed manner. One of the reviewers mentioned that the score will be raised from weak accept to strong accept. Future hardware evaluation would be highly suggested.

**Best Paper Nomination:**

No